# Creation of Annual Order Forecast for the Production of Beverage Cans—The Case Study

**Peter Kacmary *** , **Andrea Rosova** , **Marian Sofranko** , **Peter Bindzar** , **Janka Saderova** and **Jan Kovac**

Faculty of Mining, Ecology, Process Control and Geotechnologies, Technical University of Kosice, Letna 9, 04200 Kosice, Slovakia; andrea.rosova@tuke.sk (A.R.); marian.sofranko@tuke.sk (M.S.); peter.bindzar@tuke.sk (P.B.); janka.saderova@tuke.sk (J.S.); kovacjann@gmail.com (J.K.)

* Correspondence: peter.kacmary@tuke.sk; Tel.: +421-55-602-3158

**Abstract:** This article is focused on the creation of a system for forecasting of future orders for a specific beverage cans manufacturer. The problem comes from the irregular ordering of cans from different customers; not only national companies, but also from companies abroad. This causes fluctuations in production and consequently an irregular transport regime. That is why the beverage can producer demanded a forecasting system that would help to create an annual production plan. The aim is to analyze the ordering process for the last two years and on that basis to create a forecast system of the possible ordering process for the following year. This is necessary for the introduction of regularity of production, because the frequent transitions of the line to another assortment range or other surface printing requires long downtimes due to the technological setting, thus creating large losses due to inactivity of the production line. As drinking habits of final customers reflect the sale of cans, it was expected that sale data would have a seasonable character; this was proved after a brief analysis of the former data. After that, the appropriate forecasting methods were chosen. The methodology was created to combine multiple forecast results into one to increase the forecast objectivity. Forecasting is performed at three different levels: forecasting of assortment, forecasting of region sale and total forecasting of all orders. In spite of the change in market behavior in 2020, due to the pandemic situation in the first wave of the COVID-19 crisis, the sale of beverage cans is expected to stabilize and return to pre-crisis level as early as 2021. Then the forecasting system will fully meet the company's requirements.

**Keywords:** beverage cans; forecasting; PUSH manufacturing system; supply chain; inventory

## 1. Introduction

The presented case study is from a period when there was a relatively calm situation on the international markets. When the production of beverage cans concerns the food industry, it seems that the course of production and ordering of individual assortments, including the printing of cans, will be calm too. However, the reality shows the opposite. Frequent changes in the range, and especially the printing, create significant losses for downtime of production lines because they require time-consuming and technologically demanding preparations for change, testing and commissioning of the production line for a new print or product. These losses could be mitigated by sale analysis and the subsequent application of a planning model [1,2] based on forecasting methods for better estimation of future orders. It is expected that this will increase stocks, but on the other hand it will moderate the transition of lines to another range or graphics print and thus the frequency of downtime. In the case of beverages, the seasonality of the consumption of beverage cans was expected, but as there was no in-depth analysis of this market behavior, the company could not indicate to what level it is reflected in the ordered quantities of particular assortments.

The concerned company currently produces aluminum cans with volumes of 500 mL, 330 mL, 330 mL SLEEK and 250 mL. The annual capacity of the plant is 1.6 billion beverage

cans and the plant management is able to support additional capacities of production lines and thus adapt to customer requirements. These cans are delivered with a design for a specific customer requirement. Customers who regularly purchase goods are from Slovakia, Romania, the Czech Republic, southern Poland and Hungary, with the largest volumes going to Poland and Hungary. The company has signed contracts with these customers for the supply of cans lasting from two to four years [3].

This company did not have a relevant forecasting system in its planning routine and therefore the aim of this study was to design a forecasting system according to the manufacturer's visions and requirements, which were as follows:

- simplicity (easy implementation into the existing planning system);
- objectivity (relevance and reliability);
- accuracy (including the accuracy of individual methods).

Simplicity means that the new system will not require the acquisition, purchasing and implementation of any additional ready-made forecasting software. Today, there is a number of mathematical and statistical programs on the market that have a forecasting module which works on the basis of classical quantitative methods, as well as on the basis of sophisticated approaches of artificial intelligence (neural networks), but the company wanted to avoid this.

Objectivity is determined by the fact that the calculated forecast is not only a result according to the calculation of one method, but a compromise of results from several possible methods. Just the formulation of one result is the essential issue of this proposed forecast system.

The accuracy of the system is evaluated on the principle of MAPE (Mean Absolute Percentage Error), and the accuracy of individual methods is calculated. The most accurate method from the previous period is automatically given a higher weight when recalculating the resulting forecast. The values of the assigned weights are defined in such a way that the calculation of the combined forecast is not very sensitive to the value of the most accurate method.

The use of cans for preserving food and currently also beverages is not new. The history of using cans began a long time ago. In the 18th century, when military operations lasted months, sometimes years, Napoleon Bonaparte and his armies suffered from severe hunger. They had enough to eat, but it quickly spoiled. Napoleon therefore offered a reward to the one who invented the method of storing food—which eventually led to the production of the first can in 1810 [4,5].

Many things have happened since then. In the USA, the first canned beer was introduced in the world in 1935, and since then beverage cans have been an integral part of modern life. Today, more than 250 billion cans are sold annually worldwide [4,5].

Metal cans can be also considered as traditional beverage packaging material and are mainly used today for beer and soft drinks. Within the last three decades, these traditional beverage packaging materials are losing market share in comparison with PET bottles [6].

There are many questions regarding the safeness of using aluminum in food processing industry. As it is generally known, aluminum metal, mainly in the form of alloys with other metals, has many uses, including in consumer appliances, food packaging and cookware. Answers can be found in an extensive work from Scientific Opinion of the Panel published on behalf of European Food Safety Authority. The summary results from the panel noted that there are very few specific toxicological data for food additives containing aluminum. The available studies have a number of limitations and do not allow any dose-response relationships to be established. Nevertheless, there is evidence that several aluminum-containing compounds have the potential to produce neurotoxicity (demonstrated in laboratory experiments on animals) that affects the male reproductive system. In addition, after exposure to mothers, they have shown embryotoxicity, which affects the developmental nervous system of the offspring. In 1990, the Scientific Committee for Food declared the use of aluminum in the food industry to be safe. Today, there is evidence that long-term use of foods and beverages that are in direct contact with aluminum may exceed

PTWI (Provisional Tolerable Weekly Intake) values by some population groups, especially children, who regularly consume foods that include aluminum-containing additives [7].

To improve the manufacturing process from the point of environment and health, there was the new and environmentally friendly can manufacturing method introduced by authors Selles et al. using a novel pre-laminated two-layer polymer steel. It also provides a good contact surface for a food or beverage container [8].

The authors Ferrara and Plourde wrote about the constant dilemma of using refillable or non-refillable containers in case soft drink packaging in their scientific publication. The advantages and disadvantages can be debated for a long time, but the authors did not aim to defend and support repeatable packaging in the form of glass bottles, etc. Instead, they tried to address the importance of consumer heterogeneity and producers' response to this actual reality in the evaluation of various packaging regulations (for a given policy direction, etc.). Trends in many countries around the world clearly point to an annual decline in the use of returnable packaging for soft drinks. This status has been proven also in a publication from these authors [9].

The authors Chang et al. published a study to create an attractive beverage packaging based on consumers' affective requirements. It became one of the most important concerns to promote sales of beverages. The research was focused on various beverage categories in terms of container design. While their research was done to a bottled beverage, the design changes can be applied in aluminum cans as well [10].

According to another study from Brock and Williams, packaging of aluminum sheet has very good properties in several aspects related to health and environmental impact. The study shows that aluminum is better than glass from a packaging perspective [11]. The strongest parameter is the weight of the aluminum packaging, which also wins over the weight of plastic PET and HDPE packaging [12].

There is good recyclability of metal cans, including aluminum cans, but there is a need to constantly raise awareness of proper sorting and recycling, especially among children for reasons of protection of earth resources [13].

The consumption of beverages is generally influenced by various subjects. In the first place it is possible to recognize the seasonal dependence of selected types of beverages, especially beer and other carbonated beverages. This is claimed by the authors Hirche et al. [14]. Furthermore, there are various types of legislative restrictions on sales for consumption, but as Lichtman-Sadot points out, restriction on one hand triggers dominance on the other [15]. The point is that if access to unhealthy carbonated drinks for students in schools is restricted in any way, this restriction is offset by increased demand through the home environment. Currently, beverage consumption is influenced by the global pandemic COVID-19 crisis, and the study by the authors Vandenberg et al. confirms this. The research focused on the largest group of alcoholic beverages: beer. While in the first and second waves of lockdown, public beer consumption areas were closed, consumption shifted to retail food operations, which was reflected in an increase in consumption and, ultimately, the demand for packaging materials of beverage producers [16].

No other studies have been found in currently available publications, including the ones mentioned above, which would deal directly with the sale of beverage cans. Some data from the past global economic crisis have shown that lower purchasing power of consumers causes only minimal changes in the consumption of food and beverage items. In the case of food and beverages products, it was found that the demand for packaging materials grew during the COVID-19 crisis, because the final consumption of these products mainly took place in households [17–19]. Changes in consumer behavior, caused by the lockdowns in several countries around the world in 2020, will be reflected during the years 2021 and 2022, because in these years the consumption of beverage cans will be evident.

The authors of Babai et al. have dealt with the forecast and inventory performance in a two-stage supply chain by the classical method ARIMA with parameters (0,1,1) due to its attractive theoretical properties and empirical evidence in its support. The forecast was validated through the MSE (Mean Square Error) indicator, and their case study showed

that supply-chain (SC) costs between a manufacturer and a retailer can be reduced from 40% to 68% depending on other agreed strategies of performing SC [20].

The forecasting system is a multi-method model, which is mainly used for the forecasting of sale or consumption. According to the article from authors Keyno-Sadeghi, Ghaderi, Azade, et al. there is a description of the stochastic consumption of electricity forecasted by the data clustering techniques [21]. Other usage of the multi-method model came out from the Chinese automotive industry, provided by authors Zhang and Chen, where there is a kind of comparison model of similar car made [22]. Other combined systems of forecasting use so called fuzzy logic methodology [23], which totally differs from the classical statistical method [24].

In another work by Sakai et al. from SANYO Electric Co., Ltd. (Osaka, Japan) describes an original way of developing and fuzzy sales forecasting system for vending machines. This study was chosen for its similar object and it uses a combination of fuzzy logic and multiple regressive forecast models, which leads to results which are more objective and accurate. Multiple regressive forecast method was used to combine macro (main) and micro (additional) forecasts [25].

## 2. Materials and Methods

When evaluating the analysis of the data as a whole, there was an evident seasonality for ordering beverage cans of all kinds. The seasonal analysis was done by the Fourier analysis. For this reason, the Holt-Winters method and the Seasonal Indices method were considered to be suitable for forecasting in this case. Specifically, the multiplicative approach was chosen for calculating the forecast by the Holt-Winters method, because the multiplicative approach is preferred when the seasonal variations are changing proportionally to the level of the series. The other additive approach is preferred when the seasonal variations are roughly constant through the series [26].

The Holt-Winters method is based on the principle of exponential smoothing and, in addition to the level component itself, it also contains the trend and especially the seasonal component of the calculation [27]:

$$\text{Level}: \ L_t \ = \ \alpha \frac{Y_t}{S_{t-s}} + (1-\alpha)(L_{t-1} + b_{t-1}) \tag{1}$$

$$\text{Trend}: \ b_t \ = \ \beta(L_t - L_{t-1}) + (1-\beta)b_t \tag{2}$$

$$\text{Seasonal}: \ S_t \ = \ \gamma \frac{Y_t}{L_t} + (1-\gamma)S_{t-s} \tag{3}$$

$$\text{Forecast}: \ F_{t+m} \ = \ (L_t + b_t m)S_{t-s+m} \tag{4}$$

where:
Lt—level of the series,
bt—partial trend of the series,
St—seasonal component,
Ft+m—forecast for m periods ahead,
$\alpha$, $\beta$, $\gamma$—smoothing constants,
s—the length of seasonality.

Seasonal indices are ratio numbers in which the values of the time series are compared against the seasonally adjusted values (values without seasonal influence) that include only the trend (trend values are calculated, e.g., according to the linear regression). Then the average values of seasonal indices are calculated as simple arithmetic averages of seasonal indices, which relate to a certain period. These average indices must be calculated for several periods, because non-seasonal effects can be demonstrated in the examined period. Finally, the values of the average indices are multiplied by the particular values of the trend of seasonally adjusted values in the forecasted period, thus the forecast results of seasonal dependence are achieved [28].

Autoregressive integrated moving average (ARIMA) model was introduced by Box G. and Jenkins G. in early 1970s. As the name of the method discloses, it consists of two separate single models—autoregressive AR and moving average (MA). Basically, these two models can be coupled into a form of general and useful class of time series models called ARMA models. These models can be used when the data are stationary. This class of the models can be extended to non-stationary series by allowing differencing of the data series. These are called autoregressive integrated moving average (ARIMA) models. The general non-seasonal model is knowns as ARIMA (p, d, q) (parameters mean: p—order of the autoregressive part, d—degree of first differencing involved, q—order of the moving average part.) [29].

Seasonal ARIMA has additional parameters for seasonal part: Seasonal ARIMA (p, d, q) (P, D, Q) [29].

Combined forecast combines results from the above-mentioned forecast methods into one total forecast result or main forecast. The foundations of combined forecasting were laid by the authors Bates and Granger in their publication in 1969. They introduced the concept of combined forecasting as a way to increase the reliability of forecasting [30]. Since then, the forecast combination techniques have been developed and improved to the various forecast combination methods through empirical testing and simulations. The publication by the author Blatná provides an overview of the most used techniques for combining forecast results through averaging, regression or probabilistic models [31]. Armstrong discussed the number of methods to be considered in combination, concluding that, with respect to efficiency, five would be suitable. The author bases his suggestion on the exponential behavior of the combination gains. The combination of up to five forecasts reduces the number of errors, but when more than five methods are combined, gains get smaller and smaller at each addition [32]. Another publication by Hibon and Evgeniou deals with the idea of whether it makes sense to combine predictions and not just rely on the result of an individual method. The most important finding was that the choice of combined forecast always has less risk than when using only one method, which may even be inappropriately chosen [33].

Many works of the combined forecast, mentioned above, confirm the fact that a sophisticated technique of combinations does not guarantee that the overall result is more accurate or relevant. This is also confirmed by the works of the authors Zhou et al., Aras et al., Kurz-Kim, and Weng et al., who claim that even a simple combination model is sufficient to ensure a relevant result [34–39]. Most combination models are based on the principle of weighting assignment to individual methods. The combination model presented in this article is also based on this basic principle. It can be expressed by a weighted average and the higher weight is added to a method, which is more accurate in previous periods. At the beginning of this application (first forecast calculation) all methods are considered with the same accuracy and that is why all methods are allocated by the same weight (0.333) [20,25]. Each further calculation considers the weight distribution according to the proposed algorithm.

$$CF = Forecast_{HoltWinters} w_1 + Forecast_{SeasonalIndices} w_2 + Forecast_{ARIMA} w_3 \quad (5)$$

where:

CF—combined forecast,
$w_i$—particular weight of the forecast method *i*.

When creating an objective forecast of demand for beverage cans, which is later expected to be turned into specific orders, it is necessary to keep an important condition that individual customers have a long-term contract (the minimum validity is two years) [40–42]. In the event that contracts with a company for the supply of cans expire, these contracts need to be extended. If this is rejected by the customer, this situation must be immediately incorporated into the forecasting and planning system.

The company, which produces aluminum beverage cans, provided data from 2018 and 2019 sales for forecasting orders and for optimizing planning in the PUSH system strategy. The order unit represents one fully loaded truck, i.e., order amount means the number of trucks sold. Depending on the volume of the transported can (500 mL, 330 mL, 250 mL), one truck represents approximately 220,000 beverage cans. The data of individual orders were divided into three forecasting sections:

- **Section 1—forecasting of assortment:** Individual customers were divided into groups based on assortment: the group of beer producers (consumers of 500 mL beverage cans), the group of soft drink producers (consumers of 330 mL beverage cans) and the group of energy drink producers (consumers of 250 mL beverage cans). This forecast is important for the planning of the assortment, i.e., for the prediction of orders.
- **Section 2—forecasting of region sale:** Particular customers were redistributed to two regions (Poland and Hungary), where the company sent the most of trucks.
- **Section 3—overall forecast of all orders:** All customer orders are taken into account, including intercompany sales. This forecast is important for estimating the amount of input material.

## 3. Results

### 3.1. Forecasting of Assortment

For the purposes of forecasting creation, the particular assortment was classified into the following groups: beer, soft drinks and energy drinks. As was already mentioned, the seasonal nature of the data was expected. This was confirmed by Fourier analysis and subsequent calculation of Fourier periods with display of dependencies by using periodogram. The dominant cyclic component is referred to the second Fourier period (j = 2), i.e., course of two cycles during two years (one cycle per one year). In periodogram, the second Fourier period is represented in the column (second from the left) marked in red (Figure 1).

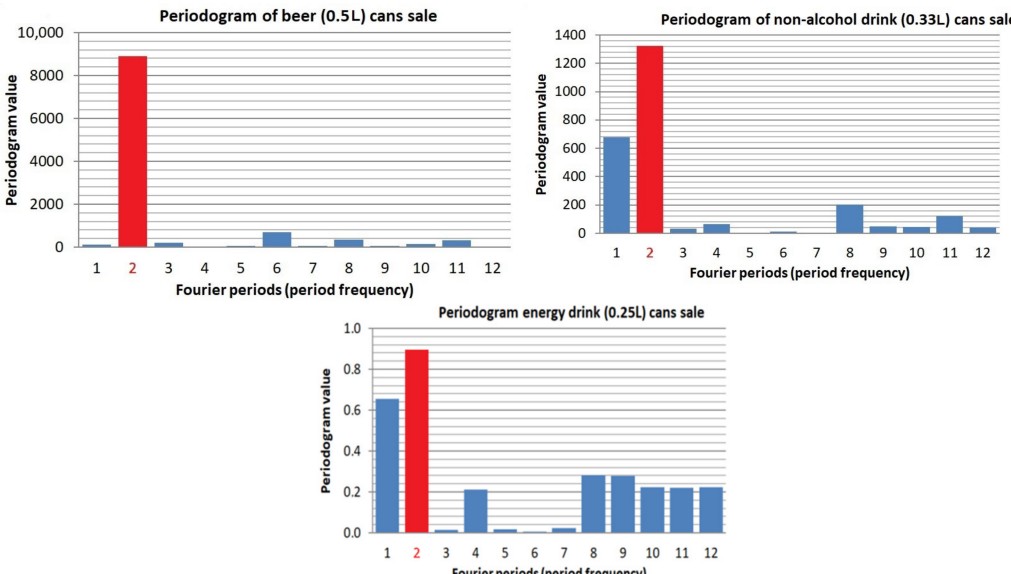

**Figure 1.** Periodograms of assortment.

In the first two months of the year (January and February) and in the last months of the year (November and December), orders for this assortment are in the smallest quantity. Demand for cans in breweries begins to rise slowly in February, i.e., at the beginning of spring, until the end of summer. Customers order cans in the largest quantities during this time compared with other months of the year. The number of orders of cans are almost double during this period (Figure 2).

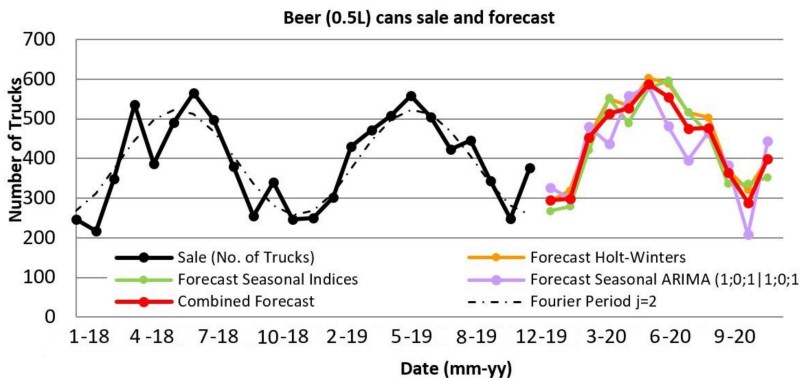

**Figure 2.** The real sale and the future forecast of 0.5 L beer beverage cans (source elaborated by authors).

There is the strongest seasonal dependence for beer drink cans (0.5 mL). This means that beer consumption has a strong dependence on weather and temperature outside (this is mainly valid for European regions).

Non-alcoholic drinks and energy drinks do not have such strong seasonal dependence, which is also proved by the forecast calculation models. However, there are still visible signs of a season (Figures 3 and 4).

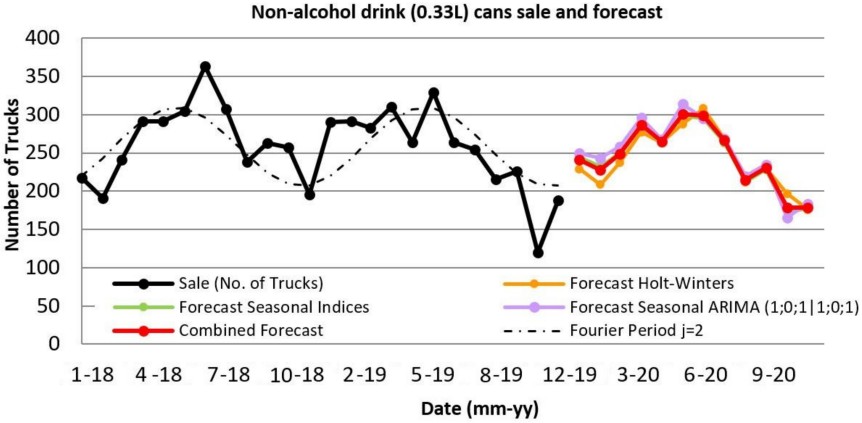

**Figure 3.** The real sale and the sale forecast of 0.33 L non-alcohol drink cans (source elaborated by authors).

Beverage cans of 0.5 L size are often referred to as beer cans and represent the largest group of produced and sold cans from the whole assortment. One of the production lines is reserved separately for this type of production in order to ensure a minimum conversion of the line to another dimension. Such adjustment may cause the line to be shut down for up to 48 h. However, idle times of the line are not completely ruled out, due to a change in graphic printing, but they are considerably less time consuming. Therefore, the production of 0.5 L cans is considered to be one of the most efficient.

### 3.2. Forecasting of Region Sale

It is possible to see the percentage ratios of volumes of goods shipped to individual countries, including Slovakia in Figure 5. The largest volume of exported cans goes to Poland, where there are nine large companies with which the producer has signed long-term contracts for the supply of beverage cans (Figure 6). A total of 6778 trucks of beverage cans were sold to this region during the years 2018–2019. In 2018, this total was 3456 trucks, while in 2019 it was 3322 trucks.

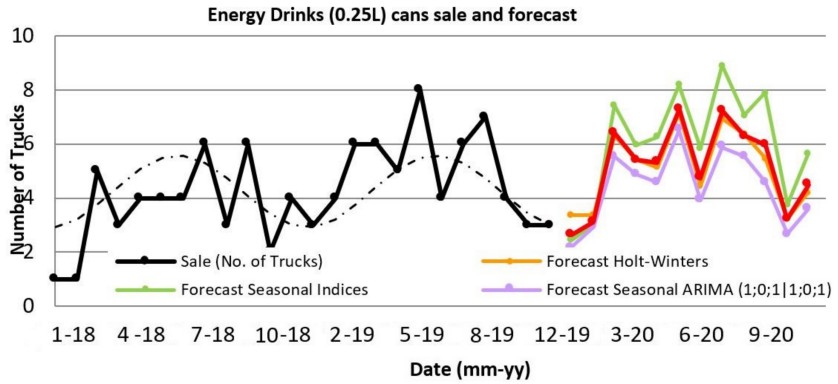

**Figure 4.** The real sale and the sale forecast of 0.25 L energy drink cans (source elaborated by authors).

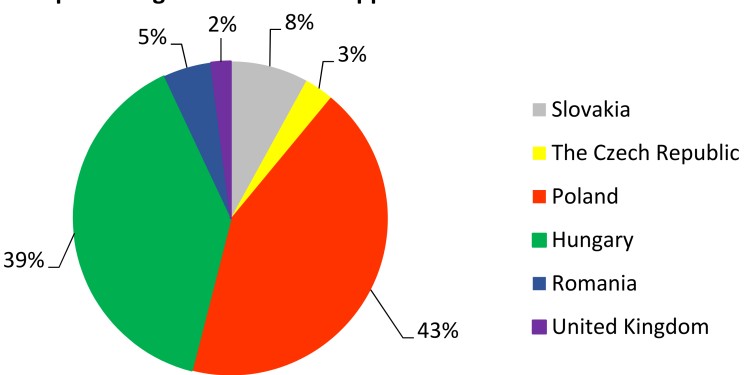

**Figure 5.** The volume of exported cans into the regions (source elaborated by authors).

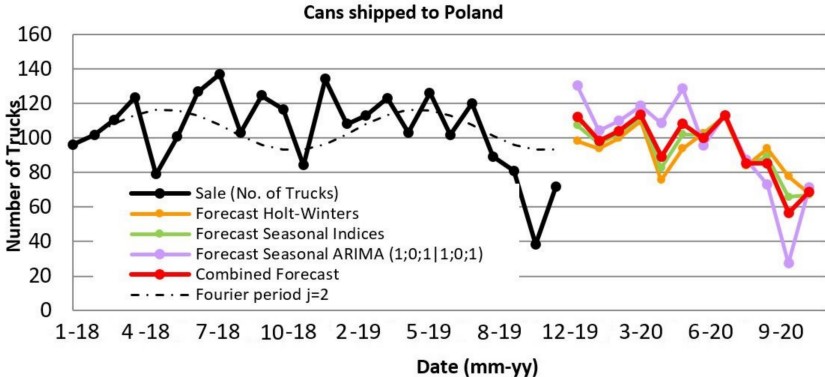

**Figure 6.** The real sale and the sale forecast of cans to Poland (source elaborated by authors).

The second largest number of cans, 6110 trucks, was sold to Hungary (Figure 7). Therefore, sales volumes to these two countries will be another subject of forecasting. Even in these sales, seasonality is reflected, so the same procedure will be used as in the case of the assortment (Figure 8).

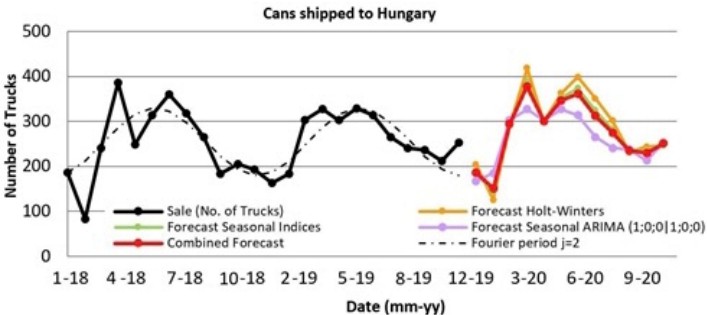

**Figure 7.** The real sale and the sale forecast of cans to Hungary (source elaborated by authors).

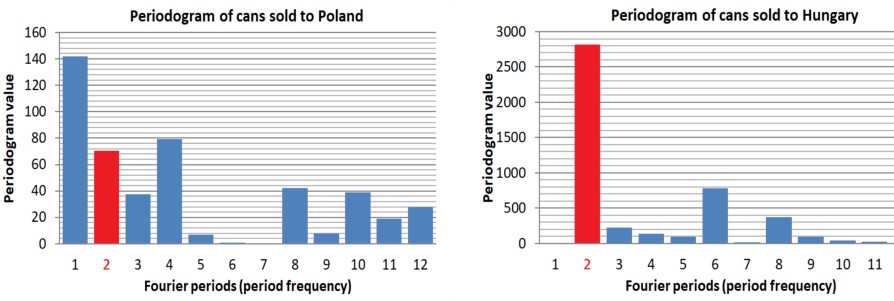

**Figure 8.** Periodograms of region sale.

Although the second Fourier period (j = 2) for the sale of cans to Poland is the only one that does not show the greatest dependence on seasonality, it can still be considered as dominant.

Beer cans (0.5 L) and cans for energy drinks (0.25 L) are mostly exported to Hungary. The consumer of cans for energy drinks is located near the border with the Slovak Republic and road distance between the producer and this consumer is less than 100 km. Therefore, this customer has signed a long-term cooperation agreement and, thanks to its short distance, the principles close to the philosophy of JiT (Just in Time) in its SCM (Supply Chain Management) are applied.

*3.3. Overall Annual Forecast of All Orders*

The overall annual forecast is important for the company mainly for ensuring of input materials. The annual production plan is decomposed into the monthly material requirements, while the seasonal dependence is also taken into account (Figure 9).

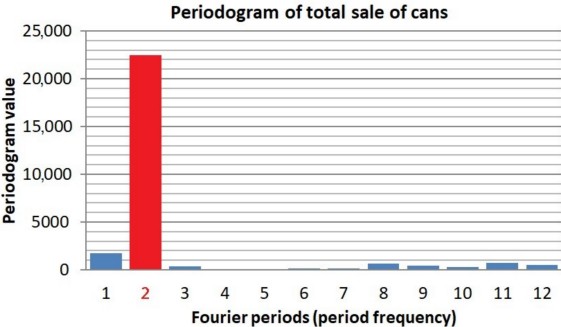

**Figure 9.** Periodogram of total sale of cans.

The following diagram (Figure 10) graphically shows all customer sales and future forecast of the company producing beverage cans, where seasonal influences are significant. Comparing between 2018 and 2019, the curve had approximately the same shape. This also proved to be a relatively stable market situation before the COVID-19 pandemic crisis from a global point of view.

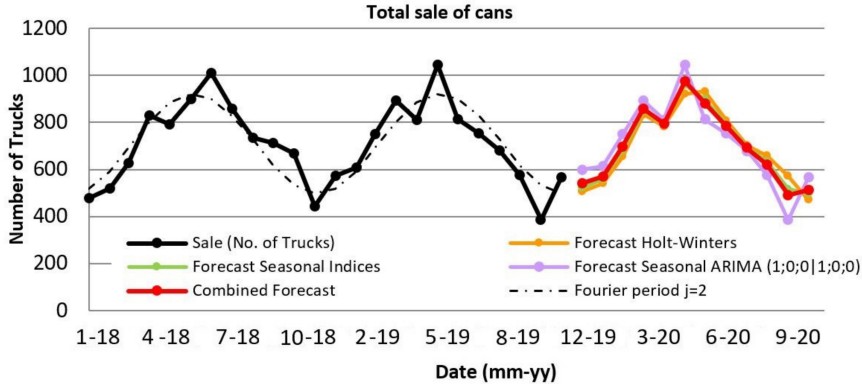

**Figure 10.** The total sale and the sale forecast of cans (source elaborated by authors).

## 4. Discussion

Three methods suitable for seasonal cases were selected for the forecast of all types of monitored sales in order to ensure higher objectivity of the forecasts. The combined forecast was created from the results of these three methods, which averages these values into one most probable result (red curves in diagrams above). When calculating the forecasts, it was assumed that market conditions or consumption of various industrial products in EU will remain relatively unchanged. However, this did not happen and in early 2020 the consequences of the pandemic crisis caused by COVID-19 began to occur. The biggest impact on the operation of the company was the strict closure of industrial enterprises during March and April 2020, as a part of anti-pandemic actions, when the production and the subsequent sale of cans fell down rapidly. Then the unfavorable development of sale stabilized and an increase in orders began to prevail during the following months. The average number of trucks transporting sold cans was 8516 during the years 2018–2019, while in 2020 it was 7809, which represents a decrease of approximately 9%. These are ultimately satisfactory results, and it corresponds to the average decline in industrial production in Slovakia, but the accuracy of the forecast was marked by the aforementioned production shutdown. The production declination and subsequent "catch-up period" (Figure 11) caused the mean absolute percentage forecast error (MAPE) to increased up to 46.5%. Other partial MAPE error indicators are listed in Table 1.

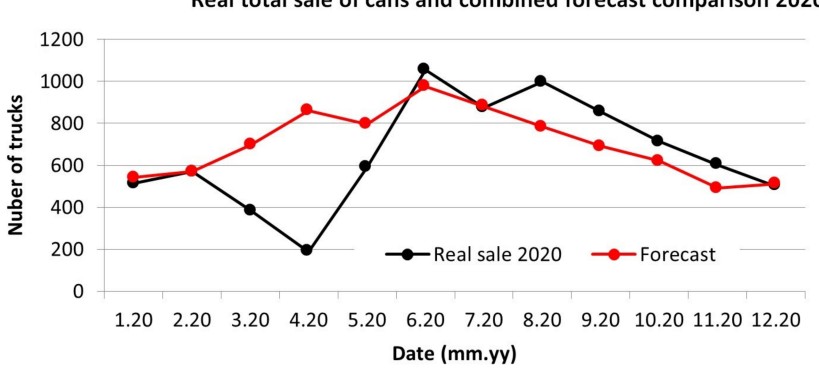

**Figure 11.** Comparison of the real total sale and forecast for year 2020 (source elaborated by authors).

**Table 1.** MAPE at particular forecasts.

| MAPE (Holt-Winters) | MAPE (Seasonal Indices) | MAPE (ARIMA) | MAPE (Comb. Forecast) |
|---|---|---|---|
| 43.87% | 45.23% | 54.18% | 46.46% |

Source: own calculation.

According to Table 1, the most accurate method was the Holt-Winters method actually. In the following calculation of the combined forecast (for the next period, e.g., for the year 2021), this method could be given a higher weight than other methods (see Formula (5)). However, this may not be the case for any other method in the future. However, the difference between MAPE and the particular forecast methods is not large and this should be maintained in the choice, or in determining of weights. The verified way in which this ratio of method accuracy is taken into account by the appropriate weight is defined by the following algorithm (Figure 12).

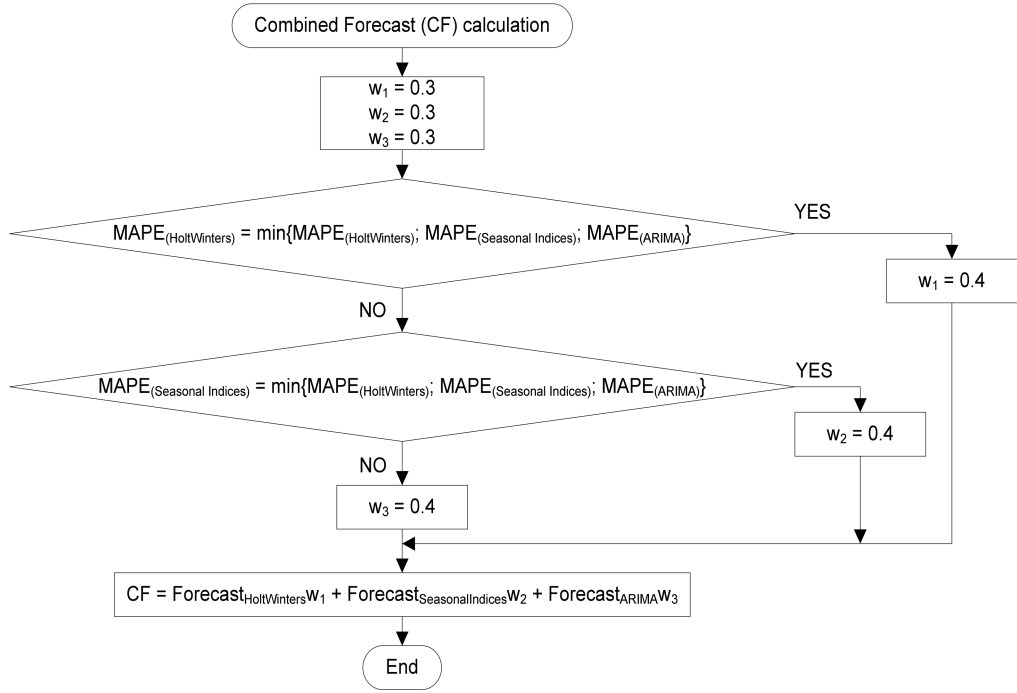

**Figure 12.** Algorithm for weight determination and CF calculation (source elaborated by authors).

The specific values of low weights ($w_i = 0.3$) and high weights ($w_i = 0.4$) have been determined heuristically and can be sensitively changed into given conditions when it is needed to give more preference to the result of a more accurate method. In this case, their sum must be equal to one.

Generally, the most accurate method from the previous period is automatically given a higher weight when recalculating the resulting forecast. The values of the assigned weights are defined in such a way that the calculation of the resulting forecast is not too sensitive to the value of the most accurate method and this is the goal. The system is unique in weight assignment, but there are still possibilities for further improvement.

## 5. Conclusions

Planners of input materials and sales, who work in the company's management, welcomed the analysis and incorporated the forecast results into their monthly and weekly plans. In this process, however, they remain irreplaceable experts who have a lot of information from the external environment (suppliers, customers, market threads or opportunities, competition, price fluctuations, rights of refusal, exchange rates, etc.) and the internal

environment (available capacities, capacity reserves, personal politics, etc.). Therefore, they have the right to change their decisions despite the forecasts provided.

The combination of quantitative and qualitative methods in the decision-making process for forecast creation also provided solutions in the past of how to get out of uncertain situations successfully [43–45]. These practices proved to be suitable for any company management, as the world has faced several years of global economic crisis since 2007, which broke the trends of market behavior.

Without the events that hit the world in 2020, the accuracy of the forecast would undoubtedly be much higher (and the MAPE indicator would be significantly lower). Apart from seasonal fluctuations in ordering of cans, which are very easy to predict due to regularity, there were no other unexpected events in the last five years. It is expected that after the end of the COVID-19 threat, the global market, not only within the EU, will be stabilized again and the prediction of future orders will again be relevant, based on the methods and techniques presented in this case-study. This model was tested with older data and its MAPE reached the value of 18.7%, which is significantly lower than the value in 2020.

As mentioned in the introduction, the effects of the pandemic crisis on the production and sale of beverage cans will become evident during the years 2021–2022. If the manufacturer is willing to cooperate in these analyses and calculations, and to provide up-to-date sale data, it will be very interesting to monitor sales characteristics, and the results should be incorporated into the calculation of more accurate forecasts. The authors of this paper will fully support not only the mentioned manufacturer, but any other companies around. Although the goal was not to incorporate the COVID-19 crisis situation into the forecasting system, it will certainly serve as inspiration for further development of such forecasting systems.

The application of the forecasting system for the company producing beverage cans is relatively fresh, because only the results from 2020 are presented in this paper. There are some other studies and applications for industrial companies, that work longer but in different conditions and principles. For example, the forecasting of products consumption having a seasonal character in ensuring steel wire ropes maintenance [46] or on the proposed forecasting system used for prediction of electro-motion spare parts demands in order to reduce stochasticity in this process [47].

**Author Contributions:** Each author (P.K., A.R., M.S., P.B., J.S. and J.K.) has equally contributed to this publication. Conceptualization, P.K. and A.R.; methodology, P.K. and A.R.; validation, P.B. and M.S.; formal analysis, P.K. and J.K.; resources, J.S. and A.R.; data curation, P.K. and A.R.; writing—original draft preparation, J.S. and P.K.; writing—review and editing, J.S. and P.K.; visualization, P.K. and P.B.; supervision, A.R. and M.S.; project administration, A.R. and M.S.; funding acquisition, A.R. and M.S. All authors have read and agreed to the published version of the manuscript.

**Funding:** This work is supported by the Scientific Grant Agency of the Ministry of Education, Science, Research, and Sport of the Slovak Republic and the Slovak Academy Sciences as part of the research project VEGA 1/0588/21; as well as by the Cultural and Educational Grant Agency of the Ministry of Education, Science, Research and Sport of the Slovak Republic and the Slovak Academy of Sciences as part of the research project KEGA 006TUKE-4/2019.

**Institutional Review Board Statement:** Not applicable.

**Informed Consent Statement:** Not applicable.

**Data Availability Statement:** The data presented in this article were obtained from the analyzed producer upon the agreement to provide data for the purpose of publishing this article only and are available on request from the corresponding author.

**Acknowledgments:** The authors would like to thank the anonymous referees for their valuable comments that improved the quality of the manuscript.

**Conflicts of Interest:** The authors declare no conflict of interest.

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
