# Peer review of "Creation of Annual Order Forecast for the Production of Beverage Cans—The Case Study"

_sustainability, doi:10.3390/su13158524_

Round 1
Reviewer 1 Report
The paper is interesting and I suggest publishing it after making the following modification.

Author Response
Comments and Suggestions for Authors
The paper is interesting and I suggest publishing it after making the following modification.
Comments from the attached PDF file:
The aim of the paper is important and very current. It is about the case study of forecasting future orders for a specific beverage can manufacturer. The paper is overall acceptable. Some minor changes should be made.
Point 1:
The Abstract should be rewritten. Please, form the abstract in the following manner. First, describe
the background of the research (1-2 sentences). Second, describe the goals of the research (1-2 sentences). Third, describe briefly (1-2 sentences) the methodology used. Fourth, describe the results and the conclusion of the research in 3-4 sentences
Response 1:
Abstract was updated according to request of the reviewer:
This article is focused on the system creation of forecasting future orders for the specific beverage can manufacturer. The problem comes from the irregular ordering of cans from different customers, not only national companies but also from companies abroad. This causes fluctuations in production and consequently an irregular transport regime. That is why, the beverage can producer demanded the forecasting system that would help to create the annual production plan. The aim was to analyze the ordering process for the last two years (years before the COVID-19 crisis) and on that basis to create make a forecast system of the possible ordering process for the following year. This is necessary for the introduction of regularity of production, because the frequent transitions of the line to another assortment range or other surface printing requires long downtimes due to the technological setting, thus creating large losses due to inactivity of the production line. Subsequently, it is necessary to adapt the warehouse regime, which will act as a buffer to compensate different material flow between production and sales. Because the drinking habits of final customers reflect the sale of cans, it was expected that sale data would have a seasonable character and it was proved after the brief analysis of the former data. After that, the appropriate forecasting methods have been chosen. There was also created the methodology for combining multiple forecast results into one to increase the forecast objectivity. The forecasting is performed in three different levels: forecasting of assortment, forecasting of region sale and total forecasting of all orders. In spite of the change in market behavior in 2020, due to the pandemic situation in the first wave of the COVID-19 crisis, the sale of beverage cans is expected to be stabilized and it returns to pre-crisis levels as early as year 2021. Then the forecasting system will fully meet the company's requirements.
Point 2:
Introduction is well written. At the end of the introduction, add two paragraphs. In the first
paragraph, explain what is the contribution of the paper, in relation to several previous papers - at
least 1-2 references from scholarly journals
Response 2:
Here are the paragraphs that were added in the introduction with three new references:
The concerned company, which produces beverage cans, did not have a relevant forecasting system in its planning routine and therefore the aim of this study was to design a forecasting system according to the manufacturer's visions and requirements, which were as follows:
- Simplicity (easy implementation into the existing planning system);
- Objectivity (relevance and reliability);
- Evaluating accuracy (including the accuracy of individual methods);
Simplicity means that it will not require the acquisition, purchasing and implementation of any additional ready-made forecasting software. Today, there are a number of mathematical and statistical programs on the market, which have a forecasting module, which works on the basis of classical quantitative methods, but as well as on the basis of sophisticated approaches of artificial intelligence (neural networks). The company wanted to avoid this.
Objectivity is given by the fact that the calculated forecast is not only a result according to the calculation of one method, but a compromise of results from several possible methods. Just the formulation of one result is the essential issue of this proposed forecast system.
The accuracy evaluation system works on the principle of MAPE (Mean Absolute Percentage Error), while the accuracy of individual methods is calculated. The most accurate method from the previous period is automatically given a higher weight when recalculating the resulting forecast. The values of the assigned weights are defined in such a way that the calculation of the resulting forecast is not very sensitive to the value of the most accurate method. The system is unique in this, but there are still possibilities to further improvement of this system.
And later in the introduction:
No studies have been found in the currently available publications, including the ones mentioned above, which would deal directly with the sale of beverage cans. Some data from the past global economic crisis have shown that lower purchasing power of consumers causes only minimal changes in the consumption of food and beverage items. In the case of food and beverages products, it was found that the demand for packaging materials is growing during the COVID-19 crisis, because the final consumption of these products takes place mainly in households [17-19]. Changes in consumer behavior, caused by the lock-downs in several countries around the world in 2020, will be reflected during the years 2021 and 2022, because in these years the consumption of beverage cans will be evident.
Point 3:
Results are well presented.
In the last sections, Discussion and Conclusion include:
Discussion why the authors found out these results and how they comply (or not) with the Literature
Review?
Response 3:
Some information about were added into the introduction part, as it is mentioned above. Here is the paragraph, that was added into the Discussion:
Generally, the most accurate method from the previous period is automatically given a higher weight when recalculating the resulting forecast. The values of the as-signed weights are defined in such a way that the calculation of the resulting forecast is not too sensitive to the value of the most accurate method and this is the goal. The system is unique in this, but there are still possibilities to further improvement of this system.
Point 4:
Limitations of the paper
Future Studies and Recommendations
Response 4:
Some information about were added into the introduction part, as it is mentioned above. Here are two paragraphs, that were added into the Conclusion:
As mentioned in the introduction, the effects of the pandemic crisis in the production and sale of beverage cans will become evident during the years 2021 – 2022. If the manufacturer is willing to cooperate in these analyses and calculations and to provide up-to-date sales data, it will be very interesting to monitor sales characteristics and then the results should be incorporated into the calculation of forecasts. The authors of this paper will fully support not only this further cooperation, but any other companies around. Although the goal was not to incorporate the COVID-19 crisis situation into the forecasting system, it will certainly serve as inspiration for further development of such forecasting systems.
The application of the forecasting system for a company producing beverage cans is relatively fresh, the results from 2020 are presented in the paper, but similar studies and applications for other companies that have been published before are already working longer and the results favorably remove the uncertainty brought by the future. For example, the forecasting of products consumption having the seasonal character in ensuring of steel wire ropes maintenance [36] or published article on the pro-posed forecasting system used for prediction of electro-motion spare parts demands in order to reduce stochasticity in this process [37].
Dear Reviewer,
Thank you for your careful reading of our manuscript and for all the valuable comments you gave us. The incorporation of the changes suggested by you surely reflects in the better quality of the new version of our manuscript.

Reviewer 2 Report
The paper titled “Creation of annual order forecast for the production of bever-2 age cans – the case study” is a relevant work in today’s scenario.
However, the following observations are made herein for further improving the quality of the paper.
- The paper is described in an abrupt way.
- Motivation of the study is not very clear. Authors should clearly mention the objectives of the study pointwise in introduction section.
- The research rationale needs to be explained with justified logic. For example, the authors have written “The aim is to analyze the ordering process for the 12 last two years (years before the COVID-19 crisis) and on that basis to make a forecast of the possible 13 ordering process for the following year” But how COVID 19 has impacted the problem area is not clear. Moreover, the impact of COVID 19 is multifaceted. Is a simple forecasting model built in pre-COVID period justified to predict post-COVID period when there is a lot of changes?
- Why the current methodology is selected? Explain clearly.
- Implications of this research is not clear.
- Result analysis needs to be more comprehensive and robust.
Author Response
Comments and Suggestions for Authors
The paper titled “Creation of annual order forecast for the production of bever-2 age cans – the case study” is a relevant work in today’s scenario.
However, the following observations are made herein for further improving the quality of the paper.
Point 1:
The paper is described in an abrupt way.
Response 1:
The paper underwent several edits after incorporating reviewers' comments. Many changes (additions in abstract, introductions, discussion and conclusion) were done throughout the paper.
English was improved by additional proofreading.
Point 2:
Motivation of the study is not very clear. Authors should clearly mention the objectives of the study pointwise in introduction section.
Response 2:
The explanation paragraphs were added into the introduction:
This company did not have a relevant forecasting system in its planning routine and therefore the aim of this study was to design the forecasting system according to the manufacturer's visions and requirements, which were as follows:
- Simplicity (easy implementation into the existing planning system);
- Objectivity (relevance and reliability);
- Evaluating accuracy (including the accuracy of individual methods).
Simplicity means that it will not require the acquisition, purchasing and implementation of any additional ready-made forecasting software. Today, there are a number of mathematical and statistical programs on the market, which have a forecasting module, which works on the basis of classical quantitative methods, but as well as on the basis of sophisticated approaches of artificial intelligence (neural networks). The company wanted to avoid this.
Objectivity is given by the fact that the calculated forecast is not only a result ac-cording to the calculation of one method, but a compromise of results from several possible methods. Just the formulation of one result is the essential issue of this pro-posed forecast system.
The accuracy evaluation system works on the principle of MAPE (Mean Absolute Percentage Error), while the accuracy of individual methods is calculated. The most accurate method from the previous period is automatically given a higher weight when recalculating the resulting forecast. The values of the assigned weights are de-fined in such a way that the calculation of the resulting forecast is not very sensitive to the value of the most accurate method. The system is unique in this, but there are still possibilities to further improvement of this system.
Point 3:
The research rationale needs to be explained with justified logic. For example, the authors have written “The aim is to analyze the ordering process for the 12 last two years (years before the COVID-19 crisis) and on that basis to make a forecast of the possible 13 ordering process for the following year” But how COVID 19 has impacted the problem area is not clear. Moreover, the impact of COVID 19 is multifaceted. Is a simple forecasting model built in pre-COVID period justified to predict post-COVID period when there is a lot of changes?
Response 3:
Unfortunately, the last years from which the forecast data came were 2018 and 2019, the years before the COVID-19 crisis. From the abstract, the information “(years before the COVID-19 crisis)” has been removed not to be confusing. During the implementation period (January / February 2020), no one knew that this COVID-19 crisis would reach the dimensions of a global pandemic situation and it would influence an industrial production in such magnificent way. In Discussion and in Conclusion, a comparison of the actual state of can sales and the forecast with relatively high MAPE values is presented, but with the stabilization of markets in 2021 or 2022, the system will show much lower MAPE values. Although the goal was not to incorporate the COVID-19 crisis situation into the forecasting system, it will certainly serve as inspiration for further development of such systems.
Point 4:
Why the current methodology is selected? Explain clearly.
Response 4:
This was mentioned in the additional text, which was added into introduction part, as well as the abstract was enhanced by the following text:
Because the drinking habits of final customers reflect the sale of cans, it was expected that sale data would have a seasonable character and it was proved after the brief analysis of the former data. After that, the appropriate forecasting methods have been chosen. There was also created the methodology for combining multiple forecast results into one to increase the forecast objectivity. The forecasting is performed in three different levels: forecasting of assortment, forecasting of region sale and total forecasting of all orders. In spite of the change in market behavior in 2020, due to the pandemic situation in the first wave of the COVID-19 crisis, the sale of beverage cans is expected to be stabilized and it returns to pre-crisis levels as early as year 2021. Then the forecasting system will fully meet the company's requirements.
Point 5:
Implications of the research is not clear.
Response 5:
Following implication was added into the discussion part:
Generally, the most accurate method from the previous period is automatically given a higher weight when recalculating the resulting forecast. The values of the as-signed weights are defined in such a way that the calculation of the resulting forecast is not too sensitive to the value of the most accurate method and this is the goal. The system is unique in this, but there are still possibilities to further improvement of this system.
Point 6:
Results analysis needs to be more comprehensive and robust.
Response 6:
Following implication was added into the conclusion part:
The application of the forecasting system for a company producing beverage cans is relatively fresh, the results from 2020 are presented in the paper, but similar studies and applications for other companies that have been published before are already working longer and the results favorably remove the uncertainty brought by the future. For example, the forecasting of products consumption having the seasonal character in ensuring of steel wire ropes maintenance [36] or published article on the pro-posed forecasting system used for prediction of electro-motion spare parts demands in order to reduce stochasticity in this process [37].
Dear Reviewer,
Thank you for your careful reading of our manuscript and for all the valuable comments you gave us. The incorporation of the changes suggested by you surely reflects in the better quality of the new version of our manuscript.

Reviewer 3 Report
In its current form, the paper looks like a case study on the application of forecasting methods for a given framework. This may be pedagogically relevant, but in terms of scholarly research, I neither see any new methodological development, nor any contextual development. It is true that COVID-19 made some difficulties with the planning horizon, yet I am not sure if this would be enough of a novel contribution to the field. What is the research gap? Based on which findings? How did you decide to close the research gap?
In addition, some of the resources are not cited properly. I would ask the authors to make a proper check of all citations/resources/references before the submission in the future.
Author Response
Comments and Suggestions for Authors
In its current form, the paper looks like a case study on the application of forecasting methods for a given framework. This may be pedagogically relevant, but in terms of scholarly research, I neither see any new methodological development, nor any contextual development. It is true that COVID-19 made some difficulties with the planning horizon, yet I am not sure if this would be enough of a novel contribution to the field. What is the research gap? Based on which findings? How did you decide to close the research gap?
Point 1:
This may be pedagogically relevant, but in terms of scholarly research, I neither see any new methodological development, nor any contextual development. It is true that COVID-19 made some difficulties with the planning horizon, yet I am not sure if this would be enough of a novel contribution to the field.
Response 1:
The paper underwent several edits after incorporating reviewers' comments. Many changes (additions in abstract, introductions, discussion and conclusion) were done throughout the paper.
English was improved by additional proofreading.
Point 2:
What is the research gap? Based on which findings? How did you decide to close the research gap?
Response 2:
The research gap is deeply explained by the added paragraphs into the introduction, discussion and conclusion parts.
Here (in the introduction part) is explanation of the goal:
The concerned company, which produces beverage cans, did not have a relevant forecasting system in its planning routine and therefore the aim of this study was to design a forecasting system according to the manufacturer's visions and requirements, which were as follows:
- Simplicity (easy implementation into the existing planning system);
- Objectivity (relevance and reliability);
- Evaluating accuracy (including the accuracy of individual methods);
Simplicity means that it will not require the acquisition, purchasing and implementation of any additional ready-made forecasting software. Today, there are a number of mathematical and statistical programs on the market, which have a forecasting module, which works on the basis of classical quantitative methods, but as well as on the basis of sophisticated approaches of artificial intelligence (neural networks). The company wanted to avoid this.
Objectivity is given by the fact that the calculated forecast is not only a result according to the calculation of one method, but a compromise of results from several possible methods. Just the formulation of one result is the essential issue of this proposed forecast system.
The accuracy evaluation system works on the principle of MAPE (Mean Absolute Percentage Error), while the accuracy of individual methods is calculated. The most accurate method from the previous period is automatically given a higher weight when recalculating the resulting forecast. The values of the assigned weights are defined in such a way that the calculation of the resulting forecast is not very sensitive to the value of the most accurate method. The system is unique in this, but there are still possibilities to further improvement of this system.
The research gap itself is explained in the discussion and conclusion parts:
Because the drinking habits of final customers reflect the sale of cans, it was expected that sale data would have a seasonable character and it was proved after the brief analysis of the former data. After that, the appropriate forecasting methods have been chosen. There was also created the methodology for combining multiple forecast results into one to increase the forecast objectivity.
Generally, the most accurate method from the previous period is automatically given a higher weight when recalculating the resulting forecast. The values of the as-signed weights are defined in such a way that the calculation of the resulting forecast is not too sensitive to the value of the most accurate method and this is the goal. The system is unique in this, but there are still possibilities to further improvement of this system.
Point 3:
In addition, some of the resources are not cited properly. I would ask the authors to make a proper check of all citations/resources/references before the submission in the future.
Response 3:
The paper was checked and mistakes that were found in the text and in the format of references were corrected. DOI number were added in the list of references, where applicable.
Dear Reviewer,
Thank you for your careful reading of our manuscript and for all the valuable comments you gave us. The incorporation of the changes suggested by you surely reflects in the better quality of the new version of our manuscript.

Round 2
Reviewer 2 Report
The authors have made significant changes. I appreciate their effort and limitations in carrying out this research. The present form of this paper may be accepted after incorporation of a minor revision. The authors need to incorporate a detailed residual analysis report in the paper.Author Response
Comments and Suggestions for Authors
The authors have made significant changes. I appreciate their effort and limitations in carrying out this research. The present form of this paper may be accepted after incorporation of a minor revision.
Point 1:
The authors need to incorporate a detailed residual analysis report in the paper.
Response 1:
The Fourier period analysis was chosen to prove the dominance of seasonality in the time series in this paper. Then periodograms which were created, except one (from data “sale to Poland”), in each time series show a dominant cyclical component corresponding to the annual cycle of sales of beverage cans. This dominant cyclic component is referred to the second Fourier period (j = 2), i.e. course of two cycles during two years (one cycle per one year). In periodograms, the second Fourier period is represented in the column (second from the left) marked in red.
This dominant second Fourier period was also added into each diagram.
Besides the analysis mentioned above, it was found that the term and creation of “combined forecast” requires more attention. More literature reviews were added to this paper.
Dear Reviewer,
Thank you for your careful reading of our manuscript and for all the valuable comments you gave us. The incorporation of the changes suggested by you surely will reflect in the better quality of the new version of our manuscript.

Reviewer 3 Report
The authors provided a new version of the manuscript, which I read in great detail. My criticism will be based on two points; presentation and content.
Regarding the presentation of the article, I noticed many incomplete sentences such as "The history of began to be written a long time ago." which -neither grammatically, nor contextually- do not make any sense to me or to the readers of Sustainability. The article was apparently not proofread as these and similar mistakes are clear to the eye of the reader.
Regarding the content of the article, I notice a very long discussion in the introductory part on the forecasting systems for canned drinks. Nevertheless, forecasting systems for any kind of tangible items to be produced by any company is nothing new. As such, you are simply applying a known-to-the-world concept to a known-to-the-world market. This is a valuable application, but I do not see any kind of research gap from the literature that requires a closure.
Author Response
Comments and Suggestions for Authors
The authors provided a new version of the manuscript, which I read in great detail. My criticism will be based on two points; presentation and content.
Point 1:
Regarding the presentation of the article, I noticed many incomplete sentences such as "The history of began to be written a long time ago." which -neither grammatically, nor contextually- do not make any sense to me or to the readers of Sustainability. The article was apparently not proofread as these and similar mistakes are clear to the eye of the reader.
Response 1:
A further check of the text of the submitted paper was performed. The subject was accidentally deleted and it was corrected in the mentioned incomplete sentence. The authors considered that it would be appropriate to bridge the topic of the previous paragraph into a new one. Therefore another sentence was added to the beginning of the paragraph, where the history of using cans is mentioned:
“The use of cans for preserving food and currently also beverages is not new. The history of using cans began a long time ago.”…
Other correction especially in new added paragraphs were checked too.
Point 2:
Regarding the content of the article, I notice a very long discussion in the introductory part on the forecasting systems for canned drinks. Nevertheless, forecasting systems for any kind of tangible items to be produced by any company is nothing new. As such, you are simply applying a known-to-the-world concept to a known-to-the-world market. This is a valuable application, but I do not see any kind of research gap from the literature that requires a closure.
Response 2:
It was found that the term and creation of “combined forecast” requires more attention. More literature reviews were added to this paper:
The foundations of combined forecasting were laid by the authors Bates and Granger in their publication in 1969. They introduced the concept of combined forecasting as a way to increase the reliability of forecasting [30]. Since then the forecast combination techniques have been developed and improved to the various forecast combination methods through empirical testing and simulations. The publication by the author Blatná provides an overview of the most used techniques for combining forecast results through averaging, regression or probabilistic models [31]. Armstrong discussed the number of methods to be considered in combination, concluding that, with respect to efficiency, five would be suitable. The author bases his suggestion on the exponential behavior of the combination gains. The combination up to five forecasts reduces the amount of errors, but when more than five methods are combined, gains get smaller and smaller at each addition [32]. Another publication by Hibon and Evgeniou deals with the idea of whether it makes sense to combine predictions and not just rely on the result of an individual method. The most important finding was: The choice of combined forecast always has less risk than when using only one method, which may even be inappropriately chosen [33].
Many works of the combined forecast, mentioned above, confirm the fact that a sophisticated technique of combinations does not guarantee that the overall result is more accurate or relevant. This is also confirmed by the works of the authors Zhou and Yang, Aras et al., KurzKim, Weng et al., who claim that even a simple combination model is sufficient to ensure a relevant result [34-39]. Most combination models are based on the principle of weighting assignment to individual methods. The combination model presented in this article is also based on this basic principle.
Added literature sources:
[30] Bates, J.M.; Granger C.W.J. The Combination of Forecasts. Operations Research Quarterly 1969, 20(4), 451-468.
[31] Blatná, D. Approaches to Combining Forecasts. In Proceedings of the 12th International Scientific Conference Applications of Mathematics and Statistics in Economy, Uherské Hradiště, The Czech Republic, 26th – 28th August 2009; Oeconomica: Praha, The Czech Republic, 2009.
[32] Armstrong, J.S. Combining forecasts: The end of the beginning or the beginning of the end? International Journal of Forecasting 1989, 5, 585-588.
[33] Hibon, M.; Evgeniou, T. To combine or not to combine: Selecting among forecasts and their combinations. International Journal of Forecasting 2005, 21, 15-24, doi:10.1016/j.ijforecast.2004.05.002.
[34] Zou H.; Yang, Y. Combining time series models for forecasting. International Journal of Forecasting 2004, 20, 69-84, doi:10.1016/S0169-2070(03)00004-9.
[35] Aras, S.; Kocakoç, İ.D.; Polat, C. Comparative study on retail sales forecasting between single and combination methods. Journal of Business Economics and Management 2017, 18(5), 803-832, doi:10.3846/16111699.2017.1367324.
[36] Kurz-Kim, J-R. Combining forecasts using optimal combination weight and generalized autoregression. Journal of Forecasting 2008, 27, 419–432, doi:10.1002/for.1069.
[37] Weng, Y.; Zhou, L.; Zhou, S.; Qi, T. Combined Forecast of China’s Steel Demand. In Liss 2014; Zhang, Z., Shen, Z.M., Zhang, J., Zhang, R., Eds.; Verlag: Springer Berlin Heidelberg, Germany, 2015, ed. 127, pp. 1673-1678, ISBN: 978-3-662-43870-1.
[38] Gaba, A.; Tsetlin, I.; Winkler, L.R. Combining Interval Forecasts. Decision Analysis 2017, 14, doi:10.1287/deca.2016.0340.
[39] Drago, C.; Lombardi, L. Combining air quality forecasts. AIP Conference Proceedings 2016, 1738, doi:10.1063/1.4952054
I dare say that there is unlimited number of ways to provide or to calculate combined forecast. The presented methodology described in algorithm is one that was used in practice.
This model was tested with older data and its MAPE reached a value of 18.7%, which is significantly lower than the value presented in the paper (reasons: the cessation of production in industrial companies at the time of lockdown).
Dear Reviewer,
Thank you for your careful reading of our manuscript and for all the valuable comments you gave us. The incorporation of the changes suggested by you surely will reflect in the better quality of the new version of our manuscript.

Round 3
Reviewer 3 Report
In its extended version, the paper can be seen as contributing to the literature. Nevertheless, I would suggest a final proofread by a native speaker.